# Induction of Browning in White Adipocytes: Fucoidan Characterization and Gold Nanoparticle Synthesis from *Undaria pinnatifida* Sporophyll Extract

**DOI:** 10.3390/md21120603

**Published:** 2023-11-23

**Authors:** Sun Young Park, Kangmin Park, Hye Mi Kang, Woo Chang Song, Jin-Woo Oh, Young-Whan Choi, Geuntae Park

**Affiliations:** 1Bio-IT Fusion Technology Research Institute, Pusan National University, Busan 46241, Republic of Korea; rkdals2477@naver.com; 2Department of Horticultural Bioscience, Pusan National University, Myrang 50463, Republic of Korea; mimi2965@naver.com (H.M.K.); ywchoi@pusan.ac.kr (Y.-W.C.); 3Department of Nanofusion Technology, Pusan National University, Busan 46241, Republic of Korea; dck3202@naver.com (W.C.S.); ojw@pusan.ac.kr (J.-W.O.)

**Keywords:** *Undaria pinnatifida* sporophyll extract, UPS-AuNPs, fucoidan, beige adipocyte

## Abstract

Seaweed extracts and their specific polysaccharides are widely known for their ability to act as reducing and capping agents during nanoparticle synthesis. Their application is highly favored in green synthesis methods, owing to their eco-friendliness, cost-effectiveness, and remarkable time and energy efficiency. In this study, fucoidan extracted from *Undaria pinnatifida* sporophyll (UPS) is introduced as a polysaccharide that effectively serves as a dual-function reducing and capping agent for the synthesis of gold nanoparticles (AuNPs). Results from various analyses indicate that AuNPs derived from UPS extract display a uniform spherical shape with an average size of 28.34 ± 1.15 nm and a zeta potential of −37.49 ± 2.13 mV, conclusively confirming the presence of Au. The FT-IR spectra distinctly revealed the characteristic fucoidan bands on the stabilized UPS-AuNPs surface. A ^1^H-NMR analysis provided additional confirmation by revealing the presence of specific fucoidan protons on the UPS-AuNPs surface. To comprehensively evaluate the impact of UPS extract, UPS-AuNPs, and fucoidan on the biological properties of adipocytes, a rigorous comparative analysis of lipid droplet formation and morphology was conducted. Our findings revealed that adipocytes treated with UPS extract, fucoidan, and UPS-AuNPs, in that order, exhibited a reduction in the total lipid droplet surface area, maximum Ferret diameter, and overall Nile red staining intensity when compared to mature white adipocytes. Furthermore, our analysis of the effects of UPS extracts, UPS-AuNPs, and fucoidan on the expression of key markers associated with white adipose tissue browning, such as UCP1, PGC1a, and PRDM16, demonstrated increased mRNA and protein expression levels in the following order: UPS-AuNPs > fucoidan > UPS extracts. Notably, the production of active mitochondria, which play a crucial role in enhancing energy expenditure in beige adipocytes, also increased in the following order: UPS-AuNPs > fucoidan > UPS extract. These findings underscore the pivotal role of UPS extract, fucoidan, and UPS-AuNPs in promoting adipocyte browning and subsequently enhancing energy expenditure.

## 1. Introduction

Obesity is a serious health concern worldwide, as acknowledged by the World Health Organization (WHO). It is distinguished by an excessive buildup of fat resulting from an imbalance between energy intake and expenditure [1]. Obesity is intimately linked to metabolic disorders such as type 2 diabetes, hypertension, dyslipidemia, cardiovascular disease, and cancer. In fact, obesity is known to contribute to a higher incidence of these ailments [2,3]. Obesity is mostly caused by poor eating habits and sedentary lifestyles, resulting from societal trends that drastically alter the overall energy balance. In addition to traditional techniques for obesity management, current treatments emphasize “browning” of white adipocytes, which increases energy expenditure via thermogenesis [4,5,6]. White, brown, and beige adipocytes are the most common types. White adipocytes store energy in the form of triglycerides. Brown adipocytes, on the other hand, create heat and spend energy, which helps to raise metabolic rates. Beige adipocytes have comparable properties and activities as brown adipocytes, and the process of changing from white to beige adipocytes is known as “fat browning” or “transdifferentiation” [7,8]. During fat browning, which is driven by stimuli, such as cold exposure, cAMP, irisin, triiodothyronine, and beta-adrenergic receptor antagonists, beige adipocytes are stimulated to express uncoupling protein 1 (UCP1). UCP1 is a mitochondrial protein that is essential for energy consumption via thermogenesis [9,10]. UCP1 contributes to non-shivering thermogenesis by creating heat on the inner mitochondrial membrane rather than producing adenosine triphosphate (ATP). Furthermore, as white adipocytes become brown, the expression levels of UCP1 and peroxisome proliferator-activated receptor gamma coactivator 1-alpha (PGC-1) increase, which correspond to particular transcription factors associated with brown adipocytes [10]. This comparable expression pattern is an important signal of the fat browning process, indicating more efficient energy use and heat generation. Furthermore, the PR domain zinc-finger protein 16 (PRDM16) interacts directly with PGC-1, controlling transcription and serving as a common marker for both brown and beige adipocytes [10,11].

Norio Taniguchi created the term “nanotechnology” in 1974 to describe the manipulation of matter at the molecular level. It spans a wide range of scientific fields, including physics, chemistry, biology, and engineering [12,13]. Nanotherapeutics, a subset of nanotechnology, has considerable potential for overcoming the limitations of traditional medication delivery systems. Targeting, biodistribution, therapeutic indices, water solubility, and oral bioavailability are among the constraints. Nanoparticles, defined as particles with at least one dimension smaller than 100 nm, are essential building elements in nanotechnology. They are classified as carbon-, metal-, ceramic-, semiconductor-, polymeric-, and lipid-based nanoparticles [14,15]. Metal nanoparticles, particularly gold nanoparticles (AuNPs), have attracted significant interest because of their excellent optical characteristics. Metallic nanoparticles have a wide range of physical and chemical characteristics that make them useful in fields such as electronics, biosensors, food, textiles, healthcare, environmental protection, and agriculture [16,17,18]. In particular, AuNPs are frequently employed because of their distinct properties, such as their high surface-area-to-volume ratio and simplicity of production. Physical and chemical synthesis methods for metal nanoparticles often require substantial energy inputs [19,20,21]. In contrast, biological approaches that involve the utilization of natural substances, such as plants, bacteria, fungi, and algae, as reducing and capping agents offer cost-effective and environmentally friendly alternatives. Green synthesis, as it is often called, is preferred because of its economic and eco-friendly nature [22,23,24].

In recent years, *Undaria pinnatifida* sporophyll (UPS) has garnered significant attention both domestically and internationally because of its diverse medicinal and therapeutic properties as well as its remarkable antioxidant capabilities [25]. UPS, a brown seaweed primarily cultivated in countries such as Japan, China, and Korea, has traditionally been used as a source of pharmaceutical and food ingredients. However, growing awareness of the health benefits associated with seaweed consumption has led to an increase in global demand [26,27]. The UPS is rich in nutrients and functional compounds, with fucoidan and fucoxanthin being the most notable constituents. Fucoidan, a yellow-brown polysaccharide extracted from seaweed, primarily comprises l-fucose and sulfated groups, along with small amounts of d-galactose, d-mannose, d-xylose, d-glucose, uronic acids, and proteins. Fucoidans exhibit a wide range of biological effects, including anticoagulant, antiviral, immune-boosting, anti-inflammatory, lipid-lowering, antioxidant, liver and kidney function enhancements, and gastrointestinal protection [28,29,30]. Previous research has provided substantial evidence supporting the mechanism by which fucoidan controls fat formation in 3T3-L1 adipocytes through the inhibition of obesity-related markers and inflammation-related cytokines [31]. Furthermore, fucoidan effectively prevents hyperglycemia after oral glucose intake in normal mice and reduces blood glucose and serum insulin levels in diabetic mice [27,32]. Numerous studies have highlighted the robust antioxidant properties of the UPS extracts and demonstrated their significant roles in mitigating oxidative damage [26,33]. These extracts serve as valuable resources for the biological synthesis of nanoparticles, which involves the reduction in metal ions using compounds derived from the UPS [34]. This study explored the potential of fucoidans obtained from UPS extracts in the synthesis of UPS-AuNPs, which serve as both reducing and capping agents. A comparative analysis of UPS extracts, UPS-AuNPs, and the mechanism of brown adipose tissue activation is presented, shedding light on the promising applications of UPS in various fields.

## 2. Results and Discussion

### 2.1. Characterization of UPS-AuNPs

The hydrodynamic diameter, zeta potential, and polydispersity index (PDI) of the UPS-AuNPs were systematically analyzed by dynamic light scattering (DLS) technology. The average hydrodynamic diameter of UPS-AuNPs was determined to be 28.34 ± 1.15 nm, as illustrated in Figure 1A. Furthermore, the zeta potential was measured at −37.49 ± 2.13 mV, providing clear evidence of the high stability of the layered structure, as depicted in Figure 1B. In addition, to validate the nanodelivery and colloidal characteristics, the PDI value was assessed. The PDI of the UPS-AuNPs was found to be 0.347. The PDI value of 0.347 indicates a moderate level of particle size distribution, suggesting some degree of heterogeneity in the UPS-AuNPs. The color change visually demonstrates the organic presence of AuNPs generated within the UPS extract, and this color change is closely related to surface plasmon resonance (SPR) characteristics. The observed SPR band in the UV-Vis spectrum had a center wavelength of 537 nm, confirming the distinctive SPR band of the AuNPs. These results align with those of a previous study [35], indicating that the SPR band for AuNPs in aqueous solutions typically falls within the 510–560 nm range (Figure 1C,D). Transmission electron microscopy (TEM) was employed for detailed analysis of the size and shape of the AuNPs. TEM images revealed a variety of nanoparticle shapes, with the majority of AuNPs exhibiting a spherical morphology, as shown in Figure 1E. The presence of well-defined lattice structures in the high-resolution TEM (HRTEM) images unequivocally demonstrates the crystalline nature of UPS-AuNPs. The particle size of the UPS-AuNPs was measured to be approximately 17.36 ± 2.49 nm, as shown in Figure 1F. The selected area electron diffraction (SAED) pattern observed in the TEM images further confirmed the crystalline nature of the UPS-AuNPs, displaying distinct circular rings corresponding to the (111), (200), (220), and (311) planes (Figure 1G). The distribution of Au within the UPS-AuNPs was confirmed by high-angle annular dark-field (HAADF) imaging, which showed red Au particles (Figure 1H). Finally, the energy dispersive X-ray (EDX) profile of the biosynthesized UPS-AuNPs exhibited prominent Au atom signals, as illustrated in Figure 1I. These findings are further supported by the presence of high-energy absorption peaks at 0.27, 2.3, and 9.6 keV, providing clear evidence of the presence of Au.

### 2.2. Analysis of Fucoidan in UPS-AuNPs

Fourier transform infrared (FT-IR) analysis (Figure 2A) was conducted to explore the molecular interactions between the UPS extract, UPS-AuNPs, and fucoidan. The FT-IR spectra of the UPS extract and UPS-AuNPs revealed similar patterns, albeit with minor shifts, likely attributed to the synthesis of AuNPs. Notably, broad peaks at approximately 3441 cm^−1^ and 3427 cm^−1^ were observed, signifying O-H stretching, and at 2928 cm^−1^ and 2915 cm^−1^, representing C-H stretching vibrations. The absorbance bands at 1633 cm^−1^ and 1628 cm^−1^ were attributed to the presence of uronic acid C=O bonds. Furthermore, low-intensity peaks at 1379 cm^−1^ and 1376 cm^−1^ were detected, indicative of C-H bending within the inner plane of the aromatic ring. The absorbance bands at approximately 1259 cm^−1^ and 1255 cm^−1^, 1021 cm^−1^ and 1017 cm^−1^, and 842 cm^−1^ and 837 cm^−1^ were associated with the stretching vibrations of S=O, C-O-C stretching, and C-O-S stretching, respectively. In the FT-IR spectrum of fucoidan, characteristic bands associated with O-H, C=O, S=O, and C-O-C stretching were observed at 3445, 1634, 1260, and 1014 cm^−1^, respectively [25]. Upon comparing the FT-IR spectra of the UPS-AuNPs and fucoidan, it became apparent that a fucoidan coating had formed on the surface of the UPS-AuNPs. The similarities and slight shifts in the spectra observed suggest the presence of various functional groups and confirm the successful establishment of a coating of the biomolecules present in the UPS extract on the surface of the UPS-AuNPs. Proton nuclear magnetic resonance (^1^H-NMR) analysis was further performed to validate the identity of the unidentified polysaccharide extracted from the stabilized outer shell of the UPS-AuNPs as fucoidan. Figure 2B presents a comparative overview of the chemical structure of fucoidan, along with the ^1^H-NMR spectra of the UPS extract, UPS-AuNPs, and fucoidan. The distinctive signals between 1.1 and 1.3 ppm in the ^1^H-NMR spectrum of fucoidan were attributed to the methyl protons (CH3) of the fucopyranose units. These signals were similarly observed in the NMR spectrum of the UPS-AuNPs, further reinforcing the presence of methyl protons. Furthermore, the resonance characteristics of H-2 to H-5 ring protons in the range of 3.2 to 4.7 ppm confirmed the presence of 3-linked α-L-fucoses in fucoidan. This characteristic was consistently evident in the NMR spectra of the UPS and UPS-AuNPs. Chemical shifts in the range of 5.1 to 5.3 ppm are ascribed to the anomeric protons (H1α) of α-linked L-fucoses and β-linked sugars (H1β). These features were uniformly observed in the NMR spectra of the UPS extract, UPS-AuNPs, and fucoidan. Moreover, a signal at 3.3 ppm in the fucoidan NMR spectrum signifies the presence of 3-linked β-D-galactose, whereas signals between 1.8 ppm indicate the existence of α-L-fucopyranose units [25,36]. These comprehensive findings provide compelling evidence that the unidentified polysaccharide extracted from the outer shell of UPS-AuNPs is fucoidan. Overall, the FT-IR and 1H-NMR analyses provide strong evidence that fucoidan is a major component of the UPS extract and that a fucoidan coating has successfully formed on the surface of the UPS-AuNPs. Further studies are warranted.

### 2.3. Effect of UPS Extract, UPS-AuNPs, and Fucoidan on Cell Viability and Lipid Droplet Morphology

Marine brown algae have played a crucial role in the Asian diet for many years, recognized for their health benefits and widely consumed worldwide, especially by individuals dealing with various health conditions [31,37]. Fucoidan, a sulfated fucose-containing polysaccharide extracted from seaweeds, has garnered attention for its multifaceted biological properties, including being an anticoagulant, antithrombotic, anticancer, anti-inflammatory, and antiviral agent. Recent studies have shed light on fucoidan’s potential in preventing obesity, showcasing its ability to reduce weight gain and curb abdominal fat accumulation in mice subjected to a high-fat diet [27,30,38]. The impact of fucoidan on obesity is intricately connected to its capacity to impede fat cell differentiation, diminish intracellular fat accumulation, and modulate lipid metabolism. Given that obesity is a primary contributor to metabolic disorders such as hypertension, atherosclerosis, type 2 diabetes, and abnormal lipid profiles, fucoidan emerges as a promising candidate for mitigating these health issues through its involvement in adipogenesis [39,40]. Furthermore, fucoidan promoted glucose uptake in fat cells, regardless of insulin application, suggesting its potential for preventing and treating diabetes. Fucoidan has received considerable attention, owing to its potential to inhibit fat accumulation. Research is underway to explore the lipid-suppressing effects of fucoidan through adipogenic reduction, which inhibits fat cell differentiation and decreases the expression of fat cell markers [30,38]. However, relatively few systematic studies have been conducted on the ability of fucoidans to induce browning in mature fat cells. In this study, a systematic analysis of the browning of mature fat cells was conducted using substances that we extracted and synthesized (UPS extract, UPS-AuNPs, and fucoidan). We used the CCK-8 assay to investigate the impact of UPS extract, UPS-AuNPs, and fucoidan on the viability of mature adipocytes. The CCK-8 assay uses a water-soluble tetrazolium salt to assess cell viability and bioreduction in the presence of electron carriers. This reaction resulted in the generation of an orange formazan dye, which was subsequently quantified by colorimetry. Within the concentration range of 50–300 μg/mL, we observed no cytotoxicity for UPS extract, UPS-AuNPs, or fucoidan. Furthermore, cell viability remained constant and did not exhibit concentration-dependent changes in this concentration range (Figure 3A). Consequently, a fixed concentration of 200 μg/mL was maintained for UPS extract, UPS-AuNPs, and fucoidan in all subsequent experiments. Adipose tissue can be categorized into three types based on the characteristics of fat cells: white, brown, and beige adipocytes. Under cold conditions, white adipocytes transform into brown or beige adipocytes via thermogenic activity. Brown adipocytes are unique in that they contain multiple small lipid droplets that form during lipolysis, which is the breakdown of large lipid droplets [10]. The collective inhibitory effects of the UPS extract, UPS-AuNPs, and fucoidan on lipid accumulation were evaluated using flow cytometry. Treatment with these compounds significantly reduced lipid accumulation compared with that in mature adipocytes (the control group). Notably, under the same concentration of 200 µg/mL, UPS-AuNPs exhibited a stronger inhibitory effect on lipid accumulation than fucoidan and UPS extract, in that order (Figure 3B). A comparative study of the effects of the UPS extract, UPS-AuNPs, and fucoidan on multilocular lipid droplets was conducted using Nile red staining analysis. In the control group, mature adipocytes formed single lipid droplets in almost all areas of nucleated adipocytes. However, when mature adipocytes transition into beige or brown adipocytes, several small lipid droplets are formed around the nucleus. Furthermore, fluorescence microscopy revealed that the control group formed large lipid droplets in differentiated adipocytes. In contrast, the groups treated with UPS extract, UPS-AuNPs, and fucoidan exhibited a decrease in the surface area and size of the lipid droplets (Figure 3C). Comparing the total lipid droplet surface area, the mature adipocyte group showed a surface area of 272.8 ± 34.1 μM^2^, whereas that treated with UPS extract, UPS-AuNPs, and fucoidan showed 184.3 ± 25.7 μM^2^, 59.7 ± 16.7 μM^2^, and 84.1 ± 28.4 μM^2^, respectively (Figure 3D). Regarding the Maximum Feret diameter, the control group exhibited a diameter of 18.0 ± 1.3 μM, whereas that treated with UPS extract, UPS-AuNPs, and fucoidan showed 12.0 ± 3.9 μM, 6.2 ± 2.6 μM, and 8.3 ± 2.9 μM, respectively (Figure 3E). Additionally, by measuring the total Nile red staining intensity using confocal microscopy, it was confirmed that adipocytes treated with UPS-AuNPs, UPS extract, and fucoidan showed a reduction in the total Nile red staining intensity compared to that in the mature adipocyte group (Figure 3F). This reduction followed the following order: UPS-AuNPs > UPS extract > fucoidan. These findings illustrate the impact of the UPS extract, UPS-AuNPs, and fucoidan on the formation of multilocular lipid droplets in brown and beige adipocytes.

### 2.4. Effects of UPS Extract, UPS-AuNPs, and Fucoidan on Mitochondrial Biogenesis and Related Gene Expression

Adipocytes, specialized cells that store neutral fats or triglycerides, are present in two distinct types of adipose tissue in mammals: white adipose tissue (WAT) and brown adipose tissue (BAT). White adipocytes primarily store energy in the form of large neutral fat molecules, occupying approximately 85–90% of the cytoplasm and pushing the nucleus and a thin layer of cytosol to the cell periphery [6,41,42]. In contrast, brown adipose tissue regulates the body temperature and possesses a higher number of mitochondria, allowing it to metabolize substantial amounts of energy substrates to generate heat. The principal distinction between these two types of adipose tissues lies in the presence or absence of UCP-1 (uncoupling protein-1) activity within the inner mitochondrial membrane of brown adipocytes. UCP1 plays a crucial role in adaptive thermogenesis by uncoupling ATP production via the catabolic pathways of lipids and carbohydrates [10,11,43]. The resulting energy is released in the form of heat, which is distributed throughout the body owing to the rich vascularization of BAT. ‘Browning’ is the process through which white adipocytes acquire characteristics similar to those of brown adipocytes and typically occurs in subcutaneous depots of white adipose tissue, where smaller adipocytes with a greater potential for differentiation are prevalent. Adrenergic stimulation and specific factors such as PGC1-α primarily drive this transformation. UCP1 and PRDM16 serve as common markers for identifying the presence of beige adipocytes within white adipose tissue [7,8,9,44]. Therefore, it is crucial to assess the effects of the UPS extract, UPS-AuNPs, and fucoidan on the augmentation of brown and beige adipocytes. To evaluate this, RT-qPCR and Western blotting analyses were conducted to measure the expression of UCP-1, PRDM16, and PGC1a. The findings revealed that the UPS extract, UPS-AuNPs, and fucoidan increased the mRNA and protein expression of UCP-1, PRDM16, and PGC1a. Specifically, at the same concentration of 200 µg/mL, UPS-AuNPs exhibited a significantly greater upregulation of UCP-1, PRDM16, and PGC1a expression compared to fucoidan and UPS extract (Figure 4A,B). Furthermore, to gain a deeper understanding of the mechanisms underlying the induction of brown adipogenesis by the UPS extract, UPS-AuNPs, and fucoidan, we conducted immunofluorescence staining to examine the expression of UCP1. Fluorescence microscopy revealed a marked increase in UCP1 expression following treatment with UPS extract, UPS-AuNPs, and fucoidan. We also evaluated the impact of UPS extract, UPS-AuNPs, and fucoidan on mitochondrial biogenesis using MitoTracker, a fluorescent probe that specifically binds to the mitochondria (shown in red). The results revealed that the red intensity in the treated adipocytes exceeded that in mature adipocytes. Interestingly, at the same concentration, UPS-AuNPs exhibited the most significant increase, followed by the UPS extract and fucoidan, indicating a strong induction of mitochondrial biogenesis. In summary, UPS extract, UPS-AuNPs, and fucoidan not only stimulated the expression of UCP-1, PRDM16, and PGC1a but also actively promoted mitochondrial biogenesis. This provides compelling evidence supporting the potential of UPS extract, UPS-AuNPs, and fucoidan as mediators of brown adipose activation.

## 3. Materials and Methods

### 3.1. Reagents

Sigma-Aldrich (Merck KGaA, Darmstadt, Germany) provided the following reagents: Fucoidan from *Undaria pinnatifida*, HAuCl43H2O, Cell Counting Kit-8 (CCK-8), a protease inhibitor, and DAPI mounting medium. Mouse 3T3-L1 preadipocytes were obtained from the American Type Culture Collection (ATCC) in Manassas, Virginia, USA. The 3T3-L1 Differentiation Kit was obtained from BioVision (Milpitas, CA, USA). The Nile red staining kit and triglyceride colorimetric test kit were obtained from Abcam (Cambridge, MA, USA). The following ingredients were obtained from Thermo Fisher Scientific Life Sciences (Waltham, MA, USA): Dulbecco’s modified Eagle’s medium/nutrient combination F-12 (DMEM/F12), fetal bovine serum (FBS), phosphate-buffered saline (PBS), penicillin, and streptomycin. They also included 8-well chamber slides, M-PERTM Mammalian Protein Extraction Reagent, Pierce ECL Western Blotting Substrate, PureLink RNA Mini Kit, high-dose cDNA reverse kit, and SYBR Green qPCR.

### 3.2. Preparation of UPS Extract

Seaweed, known for its ecological purity, was collected off the shore of Jindo Island in South Korea. To lower the salt concentration, the dried seaweed was exposed to freshwater desalination. The dried seaweed was pulverized into a fine powder using an electric mixer to achieve particle size homogeneity. To acquire particles of the lowest feasible size, a 40–50 mesh standard test sieve was employed. Ten grams of dried seaweed and 200 mL of 80% ethanol were used for the extraction process. Extraction was performed for 18 h at 100 °C. The yield was evaluated after filtering with a No. 7 filter following extraction. The UPS powder was produced in a 5 mg/mL solution and was ready for use.

### 3.3. UPS-AuNPs Synthesis

The concentration, temperature, and reaction time of the UPS extract and metal precursor were optimized for the synthesis of UPS-AuNPs. The mixture turned dark purple, signifying successful UPS-AuNPs synthesis, after adding 1 mL of the filtered extract (5 mg/mL) to 1 μL of a 1 M HAuCl4 solution and incubating for 10 min in a water bath at 60 °C. Subsequently, the UPS-AuNPs-containing tube was quickly placed in an ice bath for a 5 min chilling interval.

### 3.4. Characterization of UPS and UPS-AuNPs

The size, zeta potential, and PDI of the UPS-AuNPs were determined using a Malvern Panalytical Zetasizer Nano-ZS90 sample delivery system, which is located in Malvern, UK. The morphology and particle size distribution of UPS-AuNPs were studied using High-Resolution Transmission Electron Microscopy (HR-TEM) on a TALOS F200X apparatus (Thermo Scientific, Waltham, MA, USA) at an electron potential of 200 kV. Fourier transform infrared (FT-IR) spectra were collected using a Spectrum GX spectrometer (Perkin Elmer Inc., Boston, MA, USA). These spectra were recorded in the wavelength range of 400 to 4000 cm^−1^ and derived from samples of USP, USP-AuNPs, and fucoidan powder. The NMR signals were obtained using an Avance Neo 500 spectrometer (Bruker BioSpin, Rheinstetten, Germany). For the USP and USP-AuNPs samples, NMR signals were recorded using a 4 mm triple resonance magic angle spinning probe.

### 3.5. Mature Adipocyte Culture and Treatment

Preadipocytes of strain 3T3-L1 were grown in DMEM/F12 media supplemented with 10% FBS and 1% penicillin-streptomycin. Differentiation into mature adipocytes was stimulated according to the manufacturer’s protocol. The ability of UPS extract, UPS-AuNPs, and fucoidan to suppress adipocyte development was investigated. They were introduced 2 h before exposure to the MDI differentiation medium and maintained at a concentration of 200 g/mL for 9 days.

### 3.6. Cell Viability

For the cell viability assessment, adipocytes were seeded overnight at a density of 1 × 10^4^ cells/well in 96-well plates. After cell adhesion, different doses of UPS extract, UPS-AuNPs, and fucoidan (50, 100, 150, 200, and 300 µg/mL) were applied for 24 h. Cell viability was assessed using the CCK-8 assay kit. Following treatment, the CCK-8 working solution was added and incubated at 37 °C for 4 h. The microplate reader measured absorbance at 450 nm to determine cell viability.

### 3.7. Lipid Droplet Morphology Using Nile Red Staining

Premature adipocytes were cultured in 8-well chamber slides for Nile red staining, and mature adipocytes were generated using the procedure outlined above. After induction, mature adipocytes were rinsed with PBS, fixed in 10% formalin buffer, and stained for 0.5 h at room temperature with a Nile red solution. Images of the stained lipid droplets were acquired using a confocal microscope (LSM 800; Carl Zeiss, Jena, Germany), and the droplets were quantified using a flow cytometer (Fit NxT Flow Cytometer; Thermo Fisher Scientific).

### 3.8. Reverse Transcription-Quantitative Polymerase Chain Reaction (RT-qPCR)

We used quantitative real-time polymerase chain reaction (qRT-PCR) to determine how the UPS extract, UPS-AuNPs, and fucoidan affected the UCP-1, PRDM16, and PGC1a mRNA transcripts [43]. Cellular RNA was isolated using the PureLink RNA Mini Kit and converted into complementary DNA (cDNA) using a high-dose cDNA reverse kit according to the manufacturer’s instructions. Target transcripts were subsequently amplified using the SYBR Green qPCR Master Mix and Chromo4 (Bio-Rad, Hercules, CA, USA).

### 3.9. Western Blot Examination

With 30–50 g of protein sample loaded, the conventional Western blot technique was used. UCP1, PRDM16, and PGC1a were used as primary antibodies [43], and -actin was used as the internal control antibodies. ImageQuant TL (version 8.1; Amersham, UK) was used to quantify the grayscale values.

### 3.10. Immunofluorescence Assay

For immunofluorescence analysis, preadipocytes were seeded in an 8-well chamber slide and encouraged to develop into mature adipocytes. Following treatment, a standard immunofluorescence technique was used. The adipocytes were treated with an anti-UCP1 antibody (1:500 dilution). Finally, the coverslips were examined under a Zeiss LSM 800 confocal laser scanning microscope (Zeiss, Germany).

### 3.11. Statistical Analysis

A one-way ANOVA was used for the statistical analyses, followed by Dunnett’s multiple comparison test. A *p*-value of 0.05 was employed to ascertain statistical significance. *p*-values less than 0.05 and 0.01, respectively, are represented by the symbols “*” and “**”.

## 4. Conclusions

In summary, this study established the role of the fucoidan derived from UPS extract as a reducing and capping agent in the synthesis of UPS-AuNPs. Characterization of UPS-AuNPs using techniques such as DLS, HR-TEM, and EDX confirmed their uniform distribution, spherical morphology, and presence of Au. The FT-IR spectra of the UPS extract and UPS-AuNPs exhibited patterns and peaks akin to those of fucoidan, confirming its presence on the surface of the stabilized nanoparticles. Additionally, ^1^H-NMR spectroscopic analysis of the UPS extract and the UPS-AuNPs verified the presence of specific protons associated with fucoidan on the surface of the UPS-AuNPs. Building upon this evidence, the impact of the UPS extract, UPS-AuNPs, and fucoidan on beige adipocyte properties was investigated. The comparative study results revealed that treatment with UPS-AuNPs exerted the most pronounced effect, reducing lipid droplet size and elevating the expression of markers associated with adipocyte browning. These findings strongly suggest that UPS-AuNPs play a crucial role in promoting adipocyte browning and enhancing energy expenditure.

## Figures and Tables

**Figure 1 marinedrugs-21-00603-f001:**
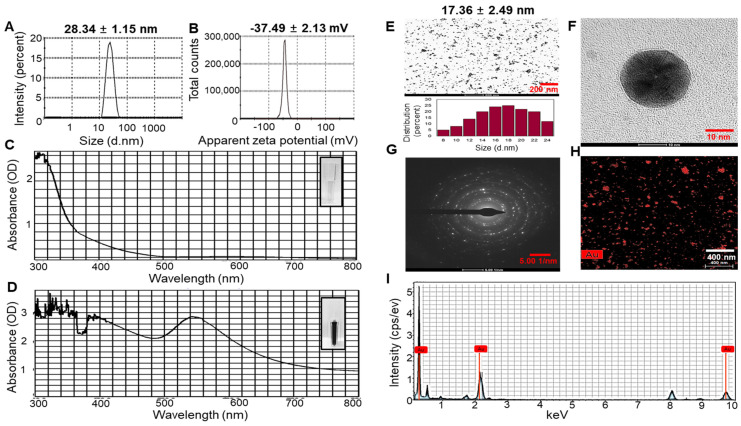
DLS and HR-TEM of UPS-AuNPs. (**A**) Hydrodynamic sizes and (**B**) zetapotential values of UPS-AuNPs. UV-Vis spectroscopy analysis of UPS extract (**C**) and green-synthesized UPS-AuNPs. (**D**,**E**) Low magnification imaging and relevant histogram analysis. (**F**) High-magnification imaging. (**G**) SAED patterns. (**H**) HAADF imaging. (**I**) EDS analysis of UPS-AuNPs.

**Figure 2 marinedrugs-21-00603-f002:**
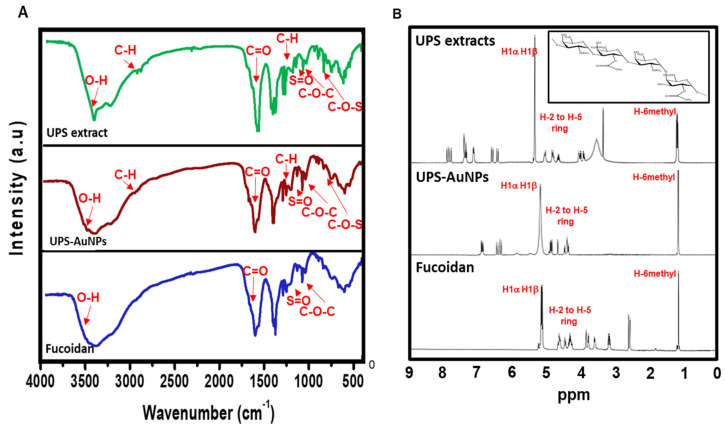
FTIR and ^1^H-NMR spectroscopy of UPS-AuNPs. (**A**) FTIR spectra of UPS extract, UPS-AuNPs, and fucoidan. (**B**) ^1^H-NMR spectrum of UPS extract, UPS-AuNPs, and fucoidan.

**Figure 3 marinedrugs-21-00603-f003:**
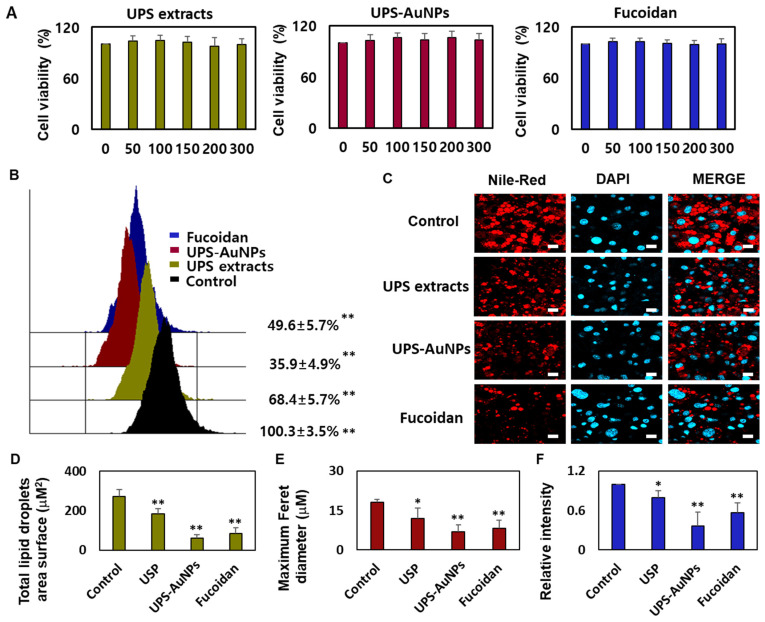
Impact of UPS extract, UPS-AuNPs, and fucoidan on cell viability and lipid droplet morphology. (**A**) Demonstrates the effects of UPS extract, UPS-AuNPs, and fucoidan (50, 100, 150, 200, and 300 µg/mL) on the vitality of adipocytes. (**B**) Nile red-O staining activity of UPS extract, UPS-AuNPs, and fucoidan (200 µg/mL), as assessed through flow cytometry. (**C**) Visual representations of the morphological changes in lipid droplets within mature adipocytes treated with UPS extract, UPS-AuNPs, and fucoidan at a concentration of 200 µg/mL (with a scale bar of 20 μm). (**D**) Collective surface area of lipid droplets in mature adipocytes upon exposure to UPS extract, UPS-AuNPs, and fucoidan at a concentration of 200 µg/mL. (**E**) Maximum Feret diameter of lipid droplets in mature adipocytes subjected to UPS extract, UPS-AuNPs, and fucoidan at a concentration of 200 µg/mL. (**F**) Overall Nile red staining intensity in mature adipocytes after treatment with UPS extract, UPS-AuNPs, and fucoidan at a concentration of 200 µg/mL using confocal microscopy. The findings are presented as the mean values along with the standard error of the mean (SEM) (*n* = 3 per group). Significance levels are indicated as * for *p* < 0.05 and ** for *p* < 0.01 concerning the comparison with the control group.

**Figure 4 marinedrugs-21-00603-f004:**
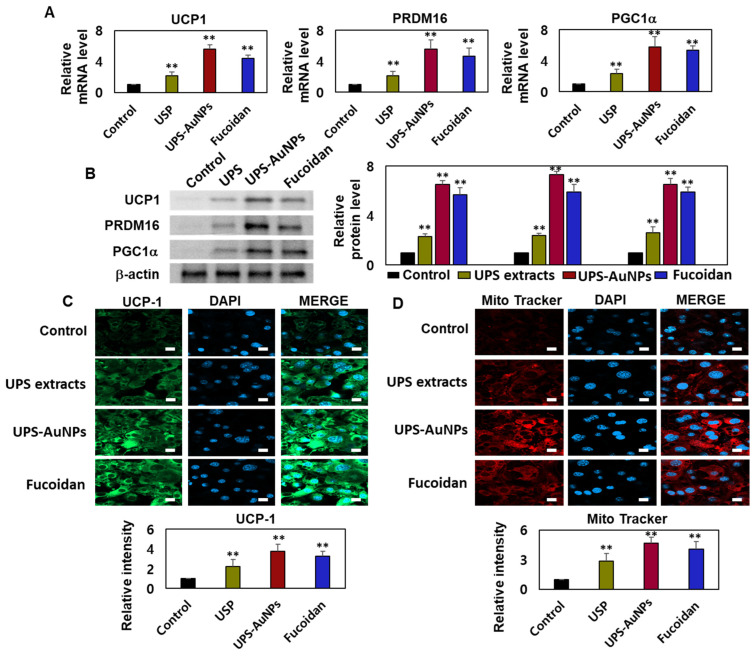
Effects of UPS extract, UPS-AuNPs, and fucoidan on mitochondrial biogenesis and related gene expression. (**A**) Transcript levels of UCP-1, PRDM16, and PGC1a were assessed using RT-qPCR. (**B**) Protein expression of UCP-1, PRDM16, and PGC1a was determined through Western blot analysis. (**C**) Quantification of UCP1 staining intensity is presented alongside representative confocal micrographs illustrating immunofluorescence staining of UCP1 and DAPI (scale bar = 20 µm). (**D**) Mitochondrial biogenesis was evaluated using MitoTracker (in red) and quantified (scale bar: 20 μm). The experiment was conducted three times, and the results are presented as the mean ± standard deviation (*n* = 3 for each group ** *p* < 0.01 vs. the control).

## Data Availability

The data supporting the findings of this study are available from the corresponding author upon reasonable request.

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
