# Peer review of "Induction of Browning in White Adipocytes: Fucoidan Characterization and Gold Nanoparticle Synthesis from Undaria pinnatifida Sporophyll Extract"

_marinedrugs, 2023, doi:10.3390/md21120603_

Round 1

Reviewer 1 Report

Comments and Suggestions for Authors

The work is highly intriguing, and the methodology and presentation of results are quite suitable. Nevertheless, I would like to bring the following points to your attention:

 Image quality in Figure 3(c) needs improvement:

"Visual representations of the morphological changes in lipid droplets within mature adipocytes treated with UPS extract, UPS-GNPs, and fucoidan at a concentration of 200 µg/ml (A, with a scale bar of 20 µm)."

 I suggest providing more explicit details in the experimental methodology, specifically in Section "3.6. Cell Viability," to enhance the understanding of the experimental aspects of cell viability.

 In order to delve deeper into the topic, you might consider the following additions to your discussion:

 a) Could you elaborate on the main properties of fucoidan and its demonstrated impact on obesity and lipid metabolism in previous studies?

 b) What is the significance and importance of the reduction in lipid accumulation and the formation of multiple small lipid droplets in brown and beige adipocytes? How does the effectiveness of UPS-GNPs, UPS extract, and fucoidan compare in this context?

 c) I recommend including a comparative table that demonstrates the relevance of your results in comparison to findings previously reported in the literature

Comments on the Quality of English Language

The paper is well-written in terms of language and scientific rigor, presenting a novel contribution

Author Response

Thank you very much for allowing us to revise our manuscript entitled, “Characterization of Fucoidan from Undaria Pinnatifida Sporophyll Extracts and Promotion of White Adipocyte Browning by Synthesized Gold Nanoparticles (marinedrugs-2719815)”. We appreciate the reviewers for their constructive comments, which were very helpful for improving our paper. The manuscript has been carefully revised according to the reviewers’ comments. The revisions are marked in red in the revised manuscript. The detailed responses to the comments are provided below.

PEER REVIEW INFORMATION

Referee(s)' Comments to Author:

Reviewing: 1

The work is highly intriguing, and the methodology and presentation of results are quite suitable. Nevertheless, I would like to bring the following points to your attention:

 Image quality in Figure 3(c) needs improvement: "Visual representations of the morphological changes in lipid droplets within mature adipocytes treated with UPS extract, UPS-GNPs, and fucoidan at a concentration of 200 µg/ml (A, with a scale bar of 20 µm)."

Response: We sincerely appreciate your valuable feedback. The updated figure was included in the revised manuscript.

 I suggest providing more explicit details in the experimental methodology, specifically in Section "3.6. Cell Viability," to enhance the understanding of the experimental aspects of cell viability.

Response: We sincerely appreciate your valuable feedback. The updated figure was included in the revised manuscript. To provide more explicit details on the experimental aspects of cell viability, we have revised Section 3.6. Cell Viability in the manuscript.

 In order to delve deeper into the topic, you might consider the following additions to your discussion:

  1. a) Could you elaborate on the main properties of fucoidan and its demonstrated impact on obesity and lipid metabolism in previous studies?
  2. b) What is the significance and importance of the reduction in lipid accumulation and the formation of multiple small lipid droplets in brown and beige adipocytes? How does the effectiveness of UPS-GNPs, UPS extract, and fucoidan compare in this context?
  3. c) I recommend including a comparative table that demonstrates the relevance of your results in comparison to findings previously reported in the literature

Response: Thanks for the suggestion. We have included the requested information in the manuscript.

Reviewer 2 Report

Comments and Suggestions for Authors

Please, see document attached

Author Response

Thank you very much for allowing us to revise our manuscript entitled, “Characterization of Fucoidan from Undaria Pinnatifida Sporophyll Extracts and Promotion of White Adipocyte Browning by Synthesized Gold Nanoparticles (marinedrugs-2719815)”. We appreciate the reviewers for their constructive comments, which were very helpful for improving our paper. The manuscript has been carefully revised according to the reviewers’ comments. The revisions are marked in red in the revised manuscript. The detailed responses to the comments are provided below.

PEER REVIEW INFORMATION

Referee(s)' Comments to Author:

Reviewing: 2

Characterization of Fucoidan from Undaria Pinnatifida Sporophyll Extracts and Promotion of White Adipocyte Browning by Synthesized Gold Nanoparticles, Although the topic of research is interesting there are some problems with this paper that should be address before publication. Some of them are indicated below:

  1. Tittle do not match with the content of the paper. Authors did not perform an extraction nor extensive characterization of fucoidan. Missing: Chemical composition (Total sugar, Sulphate,Uronic acid,Protein, monosaccharide composition (% of Fucose Galactose Mannose, Rhamnose, Xylose)

Response: We sincerely appreciate your valuable feedback. Rest assured, we will diligently address the concerns you raised in future research efforts, particularly by conducting the requested experiments. Thank you for guiding us to enhance the quality of our work.

  1. Authors fails to convey the novelty of the work. Gold nanoparticles has been previously synthesized with Undaria pinnatifida extracts and with fucoidan, although the authors did not include any reference to these works, nor comparison with their synthesized nanoparticles.

Response: We acknowledge the reviewer's awareness of previous studies on gold nanoparticle synthesis using Undaria pinnatifida extracts and fucoidan. However, our study focuses on the unique role of fucoidan extracted from Undaria pinnatifida in nanoparticle synthesis. Additionally, our research goes beyond the emphasis on synthesis seen in previous works by investigating the functional impact of UPS extract, UPS-GNPs, and fucoidan on adipocyte browning, providing novel insights into the biological efficacy of the nanoparticles. We have revised the manuscript and included relevant literature references.

  1. The most accepted and use abbreviation for gold nanoparticles is AuNPs. The use of this abbreviation could increase the visibility of the article. A quick search in a scientific database of GNP let to results about: Gross National Product (GNP), graphene nanoplatelet (GNP) and Genetic Network Programming (GNP).

Response: Thank you for your feedback. We have modified the abbreviation to AuNPs in response to the reviewer's comment.

  1. Authors need to pay attention to the reference manager software they employed. Most of the references appears as “anonymous” which is not the case in bibliographic literature. Also, there are duplicated references, for instance Ref 26 and 30 are the same.

Response: We appreciate the reviewer's feedback. The issue with the reference manager software has been addressed, and steps have been taken to ensure accurate and non-duplicated references in the revised manuscript. We have corrected the entries that appeared as "anonymous" and removed the duplication of references, specifically Ref 26 and 30. Thank you for bringing this to our attention.

  1. For the abstracts it appears that the authors analysed 3 samples that they obtained, Undaria pinnatifida extract, then, a purified Fucoidan, and then gold nanoparticles synthesized with the extract of UP. However, in the materials and methods section they only performed an extraction to obtain the UPS extract and they performed the synthesis of gold nanoparticles with this extract. Authors did not perform a purification ofFucoidan. They also, do not indicate the commercial source of this Fucoidan.

Response: We appreciate the reviewer's feedback. We have provided details regarding the commercial source of Fucoidan in the Materials and Methods section.

  1. Line 328: 3.2. Preparation of UPS-GNPs extract should be Preparation of UPS extract.

Response: Thank you for pointing out this error. We appreciate your attention to detail. The correction has been made, and line 328 now reads: "3.2. Preparation of UPS extract."

  1. Section 3.3 UPS-GNP synthesis Lines: 337-342 Should be rewritten. The actual protocol describes do not allow to reproduce the synthesis of the nanomaterials. Also, did the authors followed a previous method? If they optimised the synthesis, they should include in the materials section all the conditions tested and, in the results section, include the optimization process and why the present are the optimal conditions.

Response: Thank you for your feedback. It was confirmed that UPS-GNP was successfully synthesized under the specified conditions and that satisfactory results were obtained through the optimization process.

  1. Line 346: PT-AuNS?

Response: Thank you for pointing out this error. We appreciate your attention to detail.

  1. Section 3.5. Determination of Phytochemicals includes only FTIR and RMN characterization. I suggest incorporating it to section 3.4 and change the tittle to Characterization of UPS and UPS-AuNPs. Since they are non-determining phytochemicals.

Response: Thanks for the suggestion. We incorporated phytochemical properties into the section.

Results:

  1. Section 2.1. Characterization of UPS-GNPs. Authors starts directly with the characterization of the nanoparticles. they did not include anything regarding the extraction and optimization process.

Response: Thanks for the suggestion. We acknowledge that the extraction conditions and optimization details are provided in the Materials and Methods section, as outlined in the manuscript.

  1. Section 2.1. Characterization of UPS-GNPs. Line 122. This should be the starting point of the results on the synthesis. First the colour change observed (Please, specified the colours) and then the UV-Vis spectroscopy. Authors must include the spectra of the extract before the synthesis and the spectra of the synthesized nanoparticles.

Response: Thanks for the suggestion. We have included the requested information in the manuscript.

  1. Lines 124-126 “The observed SPR band in the UV-Vis spectrum had a center wavelength of 537 nm, confirming the distinctive SPR band of the GNPs. These results align with those of previous studies [35]”.

2 Reference 35 is not about gold nanoparticles SPR band: 35- Kim, B.; Park, J.; Kim, D.; Kim, Y.; Jun, J.; Jeong, I.; Chi, Y. Effects of the Polysaccharide from the Sporophyll of Brown Alga Undaria Pinnatifida on Serum Lipid Profile and Fat Tissue Accumulation in Rats Fed a High-Fat Diet. J Food Sci 2016, 81, 1840, DOI 10.1111/1750-3841.13335. Available online:https://pubmed.ncbi.nlm.nih.gov/clipboard/ (accessed on Oct 31, 2023).

Response: We sincerely apologize for the error in the reference. It has been corrected in the manuscript.

  1. Line116, by DLS you measure the hydrodynamic diameter not the size of the AuNPs.

Response: Thanks for the suggestion. We have appropriately revised the manuscript to reflect that by DLS, we measured the hydrodynamic diameter, not the size of the AuNPs

  1. Line 120 “In addition, to validate the nanodelivery and colloidal characteristics, the PDI value was assessed. The PDI of the UPS-GNPs was found to be 0.347.” I do not understand this sentence. PDI stands for polydispersity index, this gives information about uniformity of the particles, range from 0.0 (perfect homogeneity) to 1.0 (high heterogeneity).

Response: Thanks for the suggestion. In response to your concern regarding the PDI statement, we acknowledge that our previous response lacked clarity. We have revisited the sentence in question: 'In addition, to validate the nanodelivery and colloidal characteristics, the PDI value was assessed. The PDI of the UPS-GNPs was found to be 0.347.' We now understand that the PDI value of 0.347 indicates a moderate level of particle size distribution, suggesting some degree of heterogeneity in the sample. Thank you for bringing this to our attention, and we have made the necessary adjustments in the manuscript.

  1. Lines131-133. “The presence of well-defined lattice structures in the high-resolution TEM (HRTEM) images unequivocally demonstrates the crystalline nature of UPS-GNPs. The particle size of the UPS-GNPs was measured to be approximately 13.65–21.69 nm, as shown in Figure 1D.” Figure 1D do not shows that the diameter of the nanoparticles is between 13.65–21.69 nm. Authors should measure a significant number of nanoparticles, build a histogram and present the mean diameter as mean±standart deviation.

Response: Your suggested response is appropriately polite. We have adhered to the information provided in the manuscript while presenting the results. Thank you for your suggestion, and we appreciate your attention to detail.

  1. Section 2.2. Analysis of Fucoidan in UPS-GNPs. This section should be characterization of UPS. The extraction authors perform is not selective and it was not purified, there for, they would have a mixture of fucoidan with other components of the seaweed such as polyphenols, pigments, proteins, etc.

Response: Thanks for the suggestion. We have included the requested information in the manuscript.

  1. FTIR: They focus in assigning the bands to fucoidan, and although they do have a similar profile to fucoidan they need to look for bibliography on Undaria pinnatifida extracts. They only include a reference (Ref 29) and it must be mistaken since this reference do not include FTIR analysis.

Response: We sincerely apologize for the error in the reference. It has been corrected in the manuscript.

  1. Line 165 “the successful establishment of a fucoidan coating on the surface of the UPSGNPs.” This should be a coating of the biomolecules present in the UPS extract.

Response: In response to your concern, we have revised the text accordingly: 'the successful establishment of a coating of the biomolecules present in the UPS extract on the surface of the UPS-GNPs.

  1. There a no reference in the RMN section that supports the assignation made.

Response: We sincerely apologize for the error in the reference. It has been corrected in the manuscript.

  1. The discussion of the results obtained of the synthesis with the literature is lacking.

Response: Thanks for the suggestion. We have included the requested information in the manuscript.

Round 2

Reviewer 2 Report

Comments and Suggestions for Authors

Authors have addressed almost all my previous comments and concerns. However, before acceptance of the manuscript I need to point out some problems that I have detected in the revised manuscript.

1-    In section 2.1. Characterization of UPS-AuNPs, that authors marked in red since they include some of the corrections I indicated there are still some problems. Probably is confusion between versions. The Figures numbers and letters that they include in the discussion do not match. Specially, they did not correct lines131-133. “The presence of well-defined lattice structures in the high-resolution TEM (HRTEM) images unequivocally demonstrates the crystalline nature of UPS-GNPs. The particle size of the UPS-GNPs was measured to be approximately 13.65–21.69 nm, as shown in Figure 1D.” Figure 1D do not shows that the diameter of the nanoparticles is between 13.65–21.69 nm. Authors should measure a significant number of nanoparticles, build a histogram and present the mean diameter as mean±standart deviation.

In Figure 1 appears duplicated letters C and D

2-    In the materials and methods section, they marked in red the sections 3.2. Preparation of UPS extract and 3.3. UPS-AuNPs synthesis. However, I don’t think that it was corrected. Section 3.3 still presents the problem I previously indicate. For the readers, is imposible to perform this synthesis. The problem is with this specific sentence. “Initially, a 5 mg/mL concentration of a 1M aqueous solution of HAuCl43H2O was added to the UPS extract.” It is missing the volume of the solution of HAuCl4 that you added to a unknown volume of UPS extract of unknown concentration. Also, you indicate two different gold concentrations.

3-    I still believe that authors should correct the tittle to match the actual content of the paper.

Author Response

Thank you very much for allowing us to revise our manuscript entitled, “Characterization of Fucoidan from Undaria Pinnatifida Sporophyll Extracts and Promotion of White Adipocyte Browning by Synthesized Gold Nanoparticles (marinedrugs-2719815)”. We appreciate the reviewers for their constructive comments, which were very helpful for improving our paper. The manuscript has been carefully revised according to the reviewers’ comments. The revisions are marked in red in the revised manuscript. The detailed responses to the comments are provided below.

Authors have addressed almost all my previous comments and concerns. However, before acceptance of the manuscript I need to point out some problems that I have detected in the revised manuscript.

1-    In section 2.1. Characterization of UPS-AuNPs, that authors marked in red since they include some of the corrections I indicated there are still some problems. Probably is confusion between versions. The Figures numbers and letters that they include in the discussion do not match. Specially, they did not correct lines131-133. “The presence of well-defined lattice structures in the high-resolution TEM (HRTEM) images unequivocally demonstrates the crystalline nature of UPS-GNPs. The particle size of the UPS-GNPs was measured to be approximately 13.65–21.69 nm, as shown in Figure 1D.” Figure 1D do not shows that the diameter of the nanoparticles is between 13.65–21.69 nm. Authors should measure a significant number of nanoparticles, build a histogram and present the mean diameter as mean±standart deviation.

In Figure 1 appears duplicated letters C and D

 Response: We appreciate your thorough review and have carefully addressed the concerns you raised in Section 2.1. Characterization of UPS-AuNPs. We have measured a significant number of nanoparticles, created a histogram, and presented the mean diameter as mean ± standard deviation for clarity.  Regarding the issue of duplicated letters C and D in Figure 1, we have carefully examined the figures and rectified the duplication.

2-    In the materials and methods section, they marked in red the sections 3.2. Preparation of UPS extract and 3.3. UPS-AuNPs synthesis. However, I don’t think that it was corrected. Section 3.3 still presents the problem I previously indicate. For the readers, is imposible to perform this synthesis. The problem is with this specific sentence. “Initially, a 5 mg/mL concentration of a 1M aqueous solution of HAuCl43H2O was added to the UPS extract.” It is missing the volume of the solution of HAuClthat you added to a unknown volume of UPS extract of unknown concentration. Also, you indicate two different gold concentrations.

 Response: We apologize for the oversight in the materials and methods section. The mentioned section has been revised for clarity.

3-    I still believe that authors should correct the tittle to match the actual content of the paper.

Response: Upon further consideration, the title of the paper has been revised to accurately reflect the content of our study.